# Understanding Contrastive Learning Through the Lens of Margins

## Abstract

Contrastive learning, along with its variations, has been a highly effective self-supervised learning method across diverse domains. Contrastive learning measures the distance between representations using cosine similarity and uses cross-entropy for representation learning. Within the same framework of cosine-similarity-based representation learning, margins have played a significant role in enhancing face and speaker recognition tasks. Interestingly, despite the shared reliance on the same similarity metrics and objective functions, contrastive learning has not actively adopted margins. Furthermore, decision-boundary-based explanations are the only ones that have been used to explain the effect of margins in contrastive learning. In this work, we propose a new perspective to understand the role of margins based on gradient analysis. Based on the new perspective, we analyze how margins affect gradients of contrastive learning and separate the effect into more elemental levels. We separately analyze each and provide possible directions for improving contrastive learning. Our experimental results demonstrate that emphasizing positive samples and scaling gradients depending on positive sample angles and logits are the keys to improving the generalization performance of contrastive learning in both seen and unseen datasets, and other factors can only marginally improve performance.

## 1 Introduction

Self-supervised learning (SSL), or unsupervised learning, has attracted a lot of attention, succeeding in a range of fields Wu et al. (2018); Oord et al. (2018); Devlin et al. (2019); Dosovitskiy et al. (2021); HaoChen et al. (2021). Contrastive learning Wu et al. (2018); Oord et al. (2018) is one of the universal SSL frameworks Oord et al. (2018) not relying on domain-specific assumptions. It learns instance-level relationships between samples using the similarity function (cosine similarity) and cross-entropy. Pretraining frameworks based on contrastive learning have been applied in various fields, ranging from the image He et al. (2020); Chen et al. (2020a); Caron et al. (2020); Grill et al. (2020) to the audio domain Oord et al. (2018); Hsu et al. (2021).

As contrastive learning does, face and speaker recognition tasks share the same cosine similarity and softmax loss-based losses Zhang et al. (2019); Huang et al. (2020); Meng et al. (2021); Kim et al. (2022); Boutros et al. (2022); Desplanques et al. (2020). To improve identification accuracy, these domains usually exploit margins, which are known to increase inter-class distance and decrease intra-class distance in face and speaker recognition tasks. To be more specific, they add margins to the decision boundary by adding margins to the angles or the logits of positive samples (Eq. 4). Based on the commonality, there have been recent attempts to benefit from the characteristic of margins in contrastive learning Zhan et al. (2022); Zhang et al. (2022). These studies demonstrate that margins can improve contrastive learning-based tasks. Notwithstanding the difference between face recognition and contrastive learning, the explanation of how margins work remains solely based on the explanation from the face recognition domain.

Thus, our work starts with the following question: how do margins affect contrastive learning? It remains unclear how margins affect contrastive learning in the absence of class labels, which are commonly used in face recognition tasks to explain the effect of margins. Thus, our work aims to investigate how margins affect contrastive learning-based representation learning through gradient analysis without relying on classification- or decision-boundary-based explanations. We use the

generalized contrastive learning loss Oord et al. (2018) to incorporate margins (Sec. 3.2). Then, we analyze the gradient of the loss to identify the effect of margins on gradients (Sec. 4).

Through gradient analysis, we found that margins affect representation learning in four ways. First, it emphasizes positive samples. Second, margins reduce the gradients of distant positive samples. Third, it scales gradients by the ratio of sums of exponentiated logits without and with margins, which is affected by both the angle and logits of positive samples. Lastly, margins alleviate the slowdown effect of gradients when the estimated probability approaches the target probability. Based on the analysis, we separately explored each effect. We experimentally showed that emphasizing positive samples and scaling gradients by the ratio of sums of exponentiated logits are important for improving contrastive learning.

Contrastive learning is based on identity discrimination Wu et al. (2018), relying on the idea that representations of the same identity should cluster together while those from different identities should separate. Aside from this core idea, there are a lot of potential directions for improvement. We believe that understanding how margins affect contrastive learning can provide direction for future improvement in contrastive learning and SSL, not only helping us exploit margins in contrastive learning. Not only that, we believe our new perspective on margins could help better understand the role of margins even in other tasks, including face recognition.

To summarize, our contributions are threefolds:

• We provide a new perspective on the role of margins in cosine similarity-based representation learning through gradient analysis.

• We show that margins induce a mixture of effects, separate each effect, and experimentally validate the efficacy of each separated effect and provide its limitations.

• Our experimental results demonstrate that emphasizing positive samples and scaling gradients based on positive sample angles and logits are the keys to improving the generalization performance of contrastive learning in both seen and unseen datasets.

## 2    RELATED WORKS

### 2.1    CONTRASTIVE LEARNING

Contrastive learning (InfoNCE) Oord et al. (2018) aims to learn instance-level relationships between samples using a similarity function (cosine similarity) and cross-entropy. Contrastive learning enforces neural networks to generate close representations for positive samples (different views of the same image) and distant representations for negative samples (views from different images) Wu et al. (2018). Several InfoNCE-based SSL methods Chen et al. (2020a); Caron et al. (2020); Grill et al. (2020); He et al. (2020); Chen & He (2021) have been proposed. For example, *MoCo* He et al. (2020); Chen et al. (2020c; 2021) proposed using a teacher model for generating different latent representations of the same samples. *SimCLR* Chen et al. (2020a;b) demonstrated that augmentations play a critical role in contrastive learning frameworks. *BYOL* Grill et al. (2020), another variety of InfoNCE, proposed using only positive samples.

Several works have analyzed the properties and limitations of contrastive learning Wang & Liu (2021); Zhang et al. (2022); Wang et al. (2022); Chuang et al. (2020). Wang & Liu (2021) analyzed the role of temperature $\tau$ in contrastive learning, and found that contrastive learning focuses on nearby samples. Similarly, Zhang et al. (2022) argue that contrastive learning is robust to long-tail distribution. There are additional studies to uncover the weak spots of contrastive learning and suggest remedies. Wang et al. (2022), for example, showed that contrastive learning tends to ignore non-shared information between views, resulting in performance degradation in some downstream task. Meanwhile, Chuang et al. (2020) demonstrate that contrastive learning is biased and thus de-biasing contrastive learning loss can improve representation learning.

### 2.2    MARGIN SOFTMAX LOSS

Face and speaker recognition tasks determine whether two given samples are representations of the same identity by comparing extracted feature vectors. These tasks mainly use the same cosine-

similarity-based softmax loss as contrastive learning, the difference being using known identity information as class labels during training. Margins have been a successful method for widening inter-class, or inter-identity, distance and narrowing down intra-class distance Wang et al. (2018); Deng et al. (2019). To improve recognition performance, several variety of margins have been proposed, including adaptive Zhang et al. (2019); Huang et al. (2020); Meng et al. (2021); Kim et al. (2022) and stochastic margins Boutros et al. (2022).

There have been attempts to introduce margins also in contrastive learning (or InfoNCE) Zhan et al. (2022); Zhang et al. (2022). Zhan et al. (2022) showed that margins can enhance feature discriminability, and Zhang et al. (2022) showed that margins could be used to reduce population bias. Yet, no work has delved into gradient levels to analyze how and why margins work, and the explanations on the role of margins remain in the feature-discriminative aspects of margins. In this work, we aim to understand the margin through gradient analysis and identify the complex effects that margins have on contrastive learning.

## 3 GENERALIZED MARGINS FOR CONTRASTIVE LOSS

In this section, we generalize InfoNCE Oord et al. (2018) loss and include margins in order to analyze its effect on gradients. For consistency, we will use notations $j$ and $k$ for arbitrary indices, and notations $l$ and $h$ for positive and negative sample indices, respectively.

### 3.1 GENERALIZED INFONCE

The InfoNCE loss function can be represented as follows:

$$\tilde{q}_{ij} = \frac{\exp(sim(z_i, z_j)/\tau)}{\sum_{k \in \mathcal{X}} \exp(sim(z_i, z_k)/\tau)}, \quad \tilde{\mathcal{L}}_i = -\sum_{j \in \mathcal{X}} p_{ij} \log \tilde{q}_{ij}. \tag{1}$$

$z_i$ denotes a latent feature of input $i$. $\mathcal{X}$ denotes the set of samples in a mini-batch, and $p_{ij}$ denotes the target probability of two identities (i and j) being equal. $sim(\cdot, \cdot)$ is a similarity function. Given that cosine similarity is used as a similarity function, $\tilde{q}_{ij}$ can be rewritten as follows:

$$\theta_{ij} = \arccos(sim(z_i, z_j)), \quad \tilde{\delta}_{ij} = \cos(\theta_{ij})/\tau, \quad \tilde{q}_{ij} = \exp(\tilde{\delta}_{ij})/\sum_{k \in \mathcal{X}} \exp(\tilde{\delta}_{ik}) \tag{2}$$

$\theta_{ij}$ denotes the angle between two normalized latent features.

Other contrastive learning techniques, like BYOL Grill et al. (2020), that exclusively use positive samples cannot be covered by Eq. 1. As a result, we generalize the equation by introducing $\beta$ to the denominator of $\tilde{q}_{ij}$ as proposed in BYOL Grill et al. (2020). By rewriting the equation, we get the following equation:

$$\tilde{\mathcal{L}}_i = -\sum_{j \in \mathcal{X}} p_{ij} \log \frac{\exp(\tilde{\delta}_{ij})}{\beta \sum_{k \in \mathcal{X}} \exp(\tilde{\delta}_{ik})} = -\sum_{j \in \mathcal{X}} p_{ij} \tilde{\delta}_{ij} + \beta \sum_{j \in \mathcal{X}} p_{ij} \log \sum_{k \in \mathcal{X}} \exp(\tilde{\delta}_{ik}). \tag{3}$$

If $\beta$ is non-zero, the loss uses both positive and negative samples. Otherwise, only positive samples will be used for training. Therefore, this equation can generalize any contrastive learning-based SSL method, such as MoCo, SimCLR, and BYOL.

### 3.2 INCLUDING MARGINS IN INFONCE

There are two types of margins; angular margin $m_1$ and subtractive margin $m_2$. Angular margin $m_1$ is added to the angle between two representations $\theta_{ij}$, and subtractive margin $m_2$ is subtracted to the logits. These margins are added only to positive samples. After including margins, we can rewrite the logits and the estimated probability as follows:

$$\delta_{ij} = (\cos(\theta_{ij} + m_1 p_{ij}) - m_2 p_{ij})/\tau, \quad q_{ij} = exp(\delta_{ij})/\sum_{k \in \mathcal{X}} exp(\delta_{ik}) \tag{4}$$

Likewise, Eq. 3 can be rewritten as follows:

$$\mathcal{L}_i = -\sum_{j \in \mathcal{X}} p_{ij} \delta_{ij} + \beta \sum_{j \in \mathcal{X}} p_{ij} \log \sum_{k \in \mathcal{X}} \exp(\delta_{ik}). \tag{5}$$

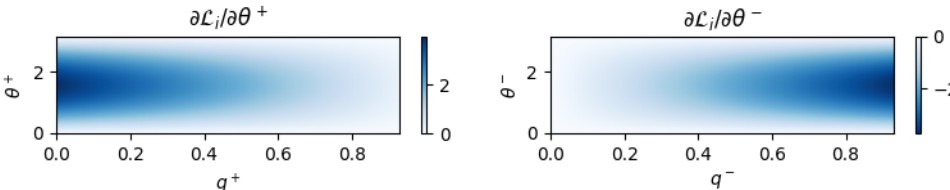

Figure 1: Gradient magnitudes of contrastive learning loss without margins (Eq. 3). $\beta$, and $\tau$ was set to 1, and 0.25, respectively. $q^+$ and $q^-$ denote the estimated probability of positive and negative samples. $\theta^+$ and $\theta^-$ refer to the angles of positive and negative samples.

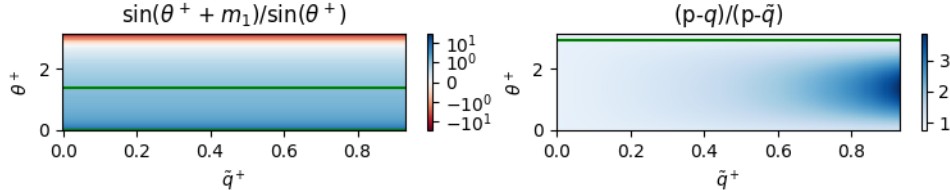

Figure 2: The gradient multipliers pertaining to the angular margin $m_1$ (Eq. 6). $\beta$, $\tau$, and $m_1$ was set to 1, 0.25, and 0.4, respectively. The left figure shows the map that applies only to positive samples, while the right figure illustrates the multiplier map that applies to both positive and negative samples. The green lines indicate where the weight is 1. Best viewed in color.

Eq. 5 can generalize to various SSL methods, including MoCo, SimCLR and BYOL by simply setting margins to zeros.

## 4   GRADIENT ANALYSIS

As Fig. 1 illustrates, the magnitude of the gradient without margins $\partial\tilde{\mathcal{L}}_i/\partial\theta_{ij}$ diminishes as the estimated probabilities $q_{ij}$ of both positive and negative samples approach their target probabilities $p_{ij}$. To examine how margins affect representation learning, we compare the derivative of Eqs. 3 and 5 with respect to the angle $\theta_{ij}$. We provide proofs of the following theorems and lemmas in the appendix.

**Theorem 4.1** *The margins $(m_1, m_2)$ scale the gradient with respect to an angle $\theta_{ij}$ by $\sin(\theta_{ij} + m_1 p_{ij})/\sin(\theta_{ij}) \cdot (p_{ij} - \beta q_{ij})/(p_{ij} - \beta\tilde{q}_{ij})$. That is,*

$$\frac{\partial\mathcal{L}_i}{\partial\theta_{ij}} = \frac{\partial\tilde{\mathcal{L}}_i}{\partial\theta_{ij}} \cdot \frac{\sin(\theta_{ij} + m_1 p_{ij})}{\sin(\theta_{ij})} \cdot \frac{p_{ij} - \beta q_{ij}}{p_{ij} - \beta\tilde{q}_{ij}}. \tag{6}$$

This shows that margins scale gradients through two terms; $\sin(\theta_{ij} + m_1 p_{ij})/\sin(\theta_{ij})$ and $(p_{ij} - \beta q_{ij})/(p_{ij} - \beta\tilde{q}_{ij})$.

**Lemma 4.2** *If $\beta$ equals one and $p_{ij}$ is either zero or one, $(p_{ij} - \beta q_{ij})/(p_{ij} - \beta\tilde{q}_{ij}) = \sum\exp(\tilde{\delta}_{ij})/\sum\exp(\delta_{ij}) = (\exp(\tilde{\delta}_{il})/q_{il})/(\exp(\delta_{il}) + \exp(\tilde{\delta}_{il})(1/q_{il} - 1)).$*

That is, it is equal to the ratio of the sums of exponentiated logits without and with margins. Based on these, we will analyze how angular margin $m_1$ and subtractive margin $m_2$ affect gradients in the following subsections.

### 4.1   ANGULAR MARGIN

As Theorem 4.1 implies, the angular margin $m_1$ multiplies the gradients by two terms: $\sin(\theta_{ij} + m_1 p_{ij})/\sin(\theta_{ij})$ and $(p_{ij} - \beta q_{ij})/(p_{ij} - \beta\tilde{q}_{ij})$. The first term $(p_{ij} - \beta q_{ij})/(p_{ij} - \beta\tilde{q}_{ij})$ emphasizes positive samples and it also de-emphasizes positive samples as the angle $\theta_{il}$ increases. The second term $\sin(\theta_{ij} + m_1 p_{ij})/\sin(\theta_{ij})$ scales gradients of both positive and negative samples. Fig. 2 visualizes these two terms. We will elaborate on them in the following sections and experimentally verify the efficacy of each component in later sections.

As shown in Fig. 2, $\sin(\theta_{ij} + m_1 p_{ij})/\sin(\theta_{ij})$ drastically increases the scale of positive sample gradients. Moreover, the gradient of a positive sample significantly diminishes as the angle $\theta_{il}$ widens. In short, the angular margin $m_1$ forces neural networks to focus on positive samples with small angles.

The second term $(p_{ij} - \beta q_{ij})/(p_{ij} - \beta \tilde{q}_{ij})$ multiplies gradients of both positive and negative samples and has several characteristics. First of all, it only relies on the angle ($\theta_{il}$ or $\theta^+$) and the estimated probability ($\tilde{q}_{il}$ or $\tilde{q}^+$) of a positive sample only. In addition, the weight increases as $\theta^+$ approaches $(\pi - m)/2$ and $\tilde{q}^+$ approaches one ($p_{il}$ or $p^+$). Unfortunately, this term cannot be expressed as a product of two arbitrary functions, $f(\theta^+)$ and $g(\tilde{q}^+)$. Therefore, we will use this ratio as it is.

## 4.2 SUBTRACTIVE MARGIN

The subtractive margin $m_2$ directly suppresses the logits of positive samples $\delta_{il}$. This attenuates the diminishing gradients as the estimated probability of a positive sample $\tilde{q}_i l$ approaches one.

**Lemma 4.3** *If $\beta$ is 1 and $p_{ij}$ is either zero or one, $\partial \mathcal{L}_i / \partial \theta_{ij}$ can be expressed as follows:*

$$\frac{\partial \mathcal{L}_i}{\partial \theta_{ij}} = \frac{\partial \tilde{\mathcal{L}}_i}{\partial \theta_{ij}} \frac{\sin(\theta_{ij} + m_1 p_{ij})}{\sin(\theta_{ij})} \frac{1}{1 - (1 - \exp((\cos(\theta_{il} + m_1) - \cos(\theta_{il}) - m_2)/\tau))\tilde{q}_{il}}, \quad (7)$$

*where $\tilde{q}_{il}$ denotes the estimated probability of a positive sample $l$ without margins.*

Given that $1/(1 - (1 - \exp((\cos(\theta_{il} + m_1) - \cos(\theta_{il}) - m_2)/\tau))\tilde{q}_{il})$ increases as $\tilde{q}_{il}$ increases, $m_2$ gives more weight as $\tilde{q}_{il}$ approaches $p_{il}$, which is one.

**Lemma 4.4** *As $m_2$ approaches infinity and $\beta$ is 1, $\lim_{m_2 \to \infty} \partial \mathcal{L}_i / \partial \theta_{il} = \sin(\theta_{il} + m_1)/\tau$.*

That is, the subtractive margin $m_2$ will make positive sample gradients independent of the estimated probabilities $\tilde{q}_{il}$ and thus of other negative samples. Since the gradient multiplier also affects negative samples, we analyze this effect in Sec. 5.5 in two ways: positive-sample-only and positive-sample-bound attenuation.

# 5 EXPERIMENTAL RESULTS

In this section, we evaluate the impact of four effects: emphasizing positive samples (Sec. 5.2), weighting positive samples differently depending on their angles (Sec. 5.3), scaling gradients by the ratio (Sec. 5.4), and attenuating the diminishing gradient effect as the estimated probability of positive sample $q_{il}$ approaches the target probability $p_{il}$ (Sec. 5.5). To this end, we conducted separate experiments on MoCov3 Chen et al. (2021), SimCLR Chen et al. (2020a), and BYOL Grill et al. (2020). We also compared with the baselines (MoCo, SimCLR, and BYOL) using linear probing (only tuning a linear layer on top of the frozen pretrained backbone model) and transfer learning (tuning the linear layer on different dataset) in Sec. 5.6. Our experiments utilized five different datasets, including CIFAR-10 Krizhevsky et al. (2009), CIFAR-100 Krizhevsky et al. (2009), STL-10 Coates et al. (2011), TinyImageNet Le & Yang (2015), and ImageNet Russakovsky et al. (2015).

## 5.1 IMPLEMENTATION DETAILS

For four datasets (CIFAR-10, CIFAR-100, STL-10, TinyImageNet), we used a modified ResNet-18 He et al. (2016) as proposed by Zheng et al. (2021). We basically followed the experimental settings from ReSSL Zheng et al. (2021). To compare with the baselines, we pretrained networks for 200 epochs and tuned only a linear layer on top of the frozen pretrained backbone model for 100 epochs (linear probing). Keep in mind that our intention is to examine the three effects on SSL methods rather than to compare the performance of different SSL methods. Therefore, we fixed the training hyper-parameters across different SSL methods.

For ImageNet, we used ResNet-50 He et al. (2016) as the backbone model, following other SSL literature He et al. (2020); Chen et al. (2020a); Grill et al. (2020). We pretrained for 100 epochs and fine-tuned a linear layer for 90 epochs, consistent with previous works Chen et al. (2021; 2020a). We did not alter any hyperparameters of the baselines, except for newly introduced hyperparameters ($s$ and $c$). Please refer to the appendix for more detailed experimental settings due to page limits.

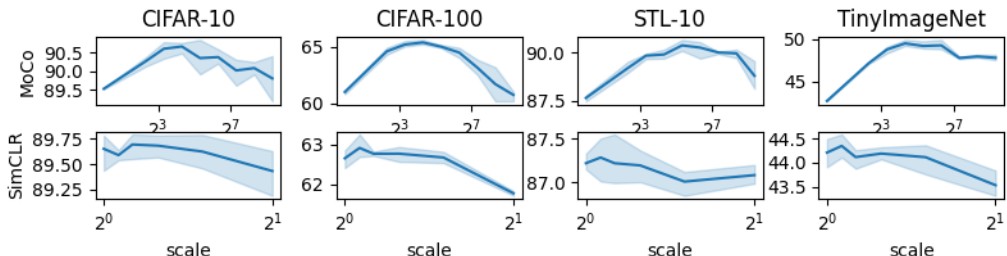

Figure 3: Top 1 accuracy when emphasizing positive samples using $s$ (Eq. 8).

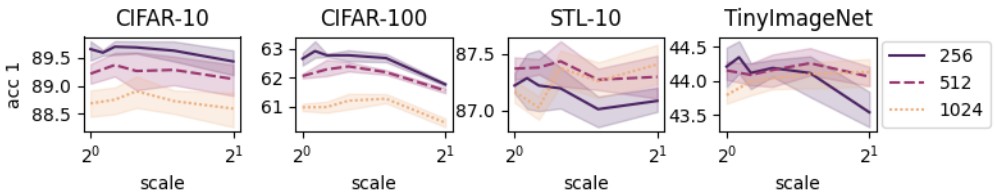

Figure 4: Relationship between the batch size and the positive sample gradient multiplier $s$ in SimCLR. The x axis is for $s$ and is in log scale.

## 5.2 EMPHASIZING POSITIVE SAMPLES

In this section, we quantitatively measure the effect of emphasizing positive samples. To this end, we introduce a new hyper-parameter $s$ to scale positive sample gradients as follows:

$$w_{ij} = (1 - p_{ij}) + s \cdot p_{ij} \tag{8}$$

$$\delta_{ij}^{scale} = \delta_{ij} w_{ij} + sg(\delta_{ij}) \cdot (1 - w_{ij}). \tag{9}$$

$w_{ij}$ denotes sample-wise weight, and $sg(\cdot)$ is the stop gradient operation. We trained neural networks by replacing $\delta_{ij}$ with $\delta_{ij}^{scale}$ in Eq. 5.

We tested the effect of emphasizing positive samples on both MoCo and SimCLR. BYOL was excluded because it only uses positive samples and thus emphasizing positive samples is the same as increasing the learning rate. Fig. 3 visualizes the results.

As the figure shows, scaling up positive sample gradients significantly improves the performance of MoCo. It shows that lowering the relative weights of negative samples can improve the performance, but if they get too small, it will rather cause performance degradation. Given that the gradients of MoCo approach those of BYOL as the scale $s$ approaches infinity (assuming the learning rate is adjusted accordingly), the presence of negative samples may be rather necessary for better SSL performance. It is also worth noting that emphasizing positive samples yields higher accuracy than using margins in the case of MoCo (Tab. 2). This implies that the performance improvement brought by emphasizing positive samples is offset by other factors, such as curvatures (Sec. 5.3).

Unlike MoCo, SimCLR does not seem to improve by scaling up positive sample gradients. However, Fig. 4 shows that as the batch size increases, the optimal $s$ increases, as does the performance gap. The larger the batch size, the more the performance curve becomes similar to that of MoCo. We believe this is related to the issue that SimCLR requires a large batch size Chen et al. (2020a).

In conclusion, while the peak and slopes are variable, it exhibits a consistent pattern across various methods and datasets. Considering that many algorithms quickly converge in a few epochs, but there is no gain in much longer training epochs, we also include the experimental results of MoCo with the training epochs of 1,000 in the appendix.

## 5.3 CURVATURE OF THE POSITIVE SAMPLE GRADIENT SCALE

In this section, we analyze the effect of weighting positive samples differently based on their angles $p_{il}$. We will refer to a weight curve, a function of the positive sample angle, as a curvature. To experiment with convex, linear, and concave curvatures, we define a curve $\gamma(x, c)$ as follows:

$$\gamma(x, c) = |(1 - x^c)^{1/c}|, \tag{10}$$

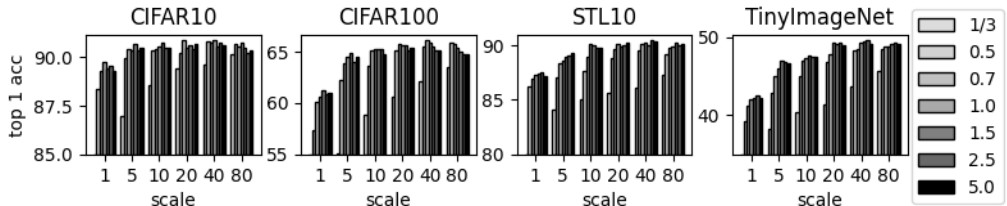

Figure 5: Controlling positive sample gradient scale curvatures using $c$ (Eq. 11). $s$ was set to one.

Figure 6: Controlling both the scale $s$ and curvature $c$ of MoCo (Eq. 11).

where $c$ is a parameter controlling the curvature, and $|\cdot|$ denotes the absolute operation. When $c$ is infinity, it becomes exactly the same as not controlling the curvature at all. To use Eq. 10 to control the diminishing rate of the positive samples gradient, we set the weights as follows;

$$w_{ij}^{dim} = (1 - p_{ij}) + \gamma(\theta_{ij}/\pi, c) \cdot s \cdot p_{ij}. \tag{11}$$

For experiments, we replaced $w_{ij}$ with $w_{ij}^{dim}$ in Eq. 9. We used three convex ($c$ is 1/3, 0.5, 0.7), three concave ($c$ is 1.5, 2.5, 5), and linear ($c$ is 1) curves.

Fig. 5 shows how different $c$ affects various SSL methods and datasets, and Fig. 6 shows the performance of MoCo as both scale $s$ and $c$ change. If $s$ is 1, the performance variation caused by $c$ is negligible, unless $c$ is extremely small. However, the performance variation caused by $c$ becomes more pronounced as $s$ increases. While it varies slightly depending on the dataset, in many cases, $c$ being close to or greater than 1 yields the optimal outcome. Furthermore, considering that the curvature due to margins is highly convex, this explains partly why using margins might not fully exploit the advantage of weighting positive samples differently based on their angles.

## 5.4 The ratio of sums of exponentiated logits

We also conducted experiments on the effect of the ratio $(p_{ij} - \beta q_{ij})/(p_{ij} - \beta\tilde{q}_{ij})$. To this end, we only multiply the gradients by the ratio without modifying the objective function. That is, we used the loss without margins (Eq. 3), while scaling gradients by the ratio. For experiments, we calculated the ratios using $m_1$ of 0.2, 0.4, 0.8, 1.6, and 3.1. Fig. 7 shows that scaling the gradients by the ratio can improve performance. Usually, the optimal margin $m_1$ lies somewhere larger than zero. But the performance curves show less congruent patterns.

## 5.5 Attenuating the diminishing gradients

As explained in Secs. 4.1 and 4.2, margins can attenuate the diminishing gradients as $q_{ij}$ approaches $p_{ij}$. Since the scale depends only on the positive sample, we ran experiments on two types of attenuation scales; type I and II. They are defined as follows:

$$w_{ij}^{I} = \sum_{k \in \mathcal{X}} \frac{p_{ik}}{1 - \alpha\tilde{q}_{ik}}, \quad w_{ij}^{II} = (1 - p_{ij}) + \frac{p_{ij}}{p_{ij} - \alpha\tilde{q}_{ij}}. \tag{12}$$

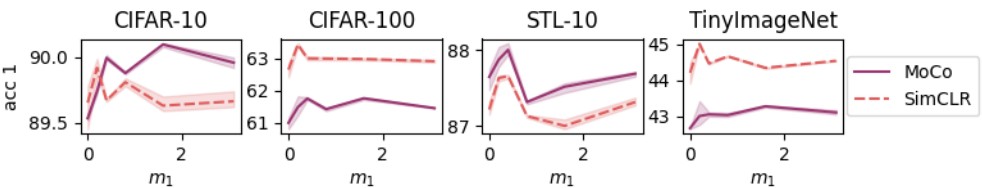

Figure 7: The 1 accuracy when scaling gradients by the ratio (Sec. 5.4).

Table 1: The effect of each gradient attenuation type and attenuation magnitude $\alpha$.

| method | $\alpha$ | CIFAR10 | | CIFAR100 | | STL10 | | TinyImageNet | |
|---|---|---|---|---|---|---|---|---|---|
| | | type I | type II | type I | type II | type I | type II | type I | type II |
| MoCo | 0 | 89.413 | | 60.956 | | 87.629 | | 42.133 | |
| | 0.25 | 89.376 | 89.713 | 60.857 | 61.217 | 87.292 | 87.455 | 42.420 | 42.460 |
| | 1 | 89.743 | 89.633 | 60.997 | 61.263 | 87.329 | 87.442 | 42.797 | 42.510 |
| SimCLR | 0 | 89.653 | | 62.655 | | 87.188 | | 44.184 | |
| | 0.25 | 89.867 | 89.557 | 62.407 | 62.763 | 87.200 | 87.042 | 44.206 | 44.197 |
| | 1 | 89.735 | 89.767 | 62.840 | 62.800 | 87.300 | 87.317 | 44.260 | 44.260 |

Table 2: Linear probing. *pos.* denotes emphasizing positive samples and *curv.* denotes controlling the curvature of positive gradient scales.

| | CIFAR10 | CIFAR100 | STL10 | TinyImageNet |
|---|---|---|---|---|
| MoCo | $89.413 \pm 0.109$ | $60.956 \pm 0.167$ | $87.629 \pm 0.175$ | $42.133 \pm 0.225$ |
| *+ margins* | $89.840 \pm 0.105$ | $61.796 \pm 0.254$ | $87.688 \pm 0.275$ | $42.860 \pm 0.280$ |
| *+ pos.* | $90.663 \pm 0.159$ | $65.413 \pm 0.205$ | $90.357 \pm 0.398$ | $48.273 \pm 0.405$ |
| *+ pos. & curv.* | $\mathbf{90.865 \pm 0.025}$ | $\mathbf{66.095 \pm 0.295}$ | $\mathbf{90.361 \pm 0.057}$ | $\mathbf{48.503 \pm 0.270}$ |
| *+ ratio.* | $90.100 \pm 0.017$ | $61.767 \pm 0.040$ | $88.000 \pm 0.078$ | $43.283 \pm 0.031$ |
| SimCLR | $89.653 \pm 0.207$ | $62.655 \pm 0.277$ | $87.188 \pm 0.145$ | $44.184 \pm 0.220$ |
| *+ margins* | $\mathbf{90.447 \pm 0.191}$ | $\mathbf{63.507 \pm 0.414}$ | $87.430 \pm 0.220$ | $44.593 \pm 0.410$ |
| *+ pos.* | $89.695 \pm 0.150$ | $62.917 \pm 0.302$ | $87.284 \pm 0.248$ | $44.348 \pm 0.295$ |
| *+ pos. & curv.* | $90.190 \pm 0.067$ | $63.503 \pm 0.279$ | $87.346 \pm 0.156$ | $44.376 \pm 0.201$ |
| *+ ratio.* | $89.920 \pm 0.066$ | $63.437 \pm 0.045$ | $\mathbf{87.658 \pm 0.029}$ | $\mathbf{45.010 \pm 0.040}$ |
| BYOL | $90.283 \pm 0.109$ | $61.006 \pm 0.140$ | $87.546 \pm 0.668$ | $41.846 \pm 0.097$ |
| *+ curv.* | $\mathbf{90.485 \pm 0.085}$ | $\mathbf{61.170 \pm 0.340}$ | $\mathbf{88.051 \pm 0.230}$ | $\mathbf{43.332 \pm 0.564}$ |

For type I, we multiply $1/(1 - \alpha\tilde{q}_{il})$ to gradients to both positive and negative samples, as $m_2$ does (Eq. 7). If $\alpha$ equals $1 - \exp((\cos(\theta_{il} + m_1) - \cos(\theta_{il}) - m_2)/\tau)$, type I equals using subtractive margin $m_2$ (Eq. 7). For type II, we only scale gradients of positive samples only.

Tab. 1 shows the performance as $\alpha$ changes. We exempt BYOL because $q_{ij}$ cannot exist without negative samples. As the table shows, attenuating positive gradients as $q_{il}$ approaches $p_{il}$ does not significantly improve performance in both cases (types I and II). This might be related to the fact that many face recognition methods use only the angular margin $m_1$.

## 5.6 COMPARISON WITH BASELINES

In this section, we compare the baselines with and without margins as well as three other components; emphasizing positive samples (in short *pos.*), weighting positive samples differently (abbreviated as *curv.*), and scaling by ratios (in short *ratio.*). We tested each model in two different settings; linear probing (Tab. 2) and transfer learning (Tab. 3). Experimental details, including exact values of $s$ and $c$, are addressed in the appendix.

Table 4: Top 1 accuracy of ResNet-50 when linear probing on ImageNet.

| | Epoch | ImageNet |
|---|---|---|
| MoCov3 | 100 | 68.9 |
| *+ pos. & curv.* | 100 | **70.9** |
| SimCLR | 100 | 64.7 |
| *+ pos. & curv.* | 100 | **65.7** |

Table 3: Transfer learning. While freezing the pretrained backbone model, only the linear layer was tuned to the target dataset. It is expressed in the form of "pretrained dataset $\rightarrow$ target dataset".

|  | CIFAR100 $\rightarrow$ CIFAR10 | CIFAR10 $\rightarrow$ CIFAR100 | TinyImageNet $\rightarrow$ STL10 | STL10 $\rightarrow$ TinyImageNet |
|---|---|---|---|---|
| MoCo | $75.417 \pm 0.241$ | $43.590 \pm 0.225$ | $72.013 \pm 0.250$ | $31.140 \pm 0.130$ |
|     + *margins* | $75.380 \pm 0.418$ | $43.190 \pm 0.255$ | $72.154 \pm 0.654$ | $30.543 \pm 0.160$ |
|     + *pos.* | $\mathbf{78.847 \pm 0.244}$ | $\mathbf{55.113 \pm 0.434}$ | $78.100 \pm 0.185$ | $\mathbf{41.825 \pm 0.431}$ |
|     + *pos. & curv.* | $78.600 \pm 0.042$ | $52.940 \pm 0.750$ | $\mathbf{78.734 \pm 0.040}$ | $41.777 \pm 0.690$ |
|     + *ratio.* | $76.027 \pm 0.074$ | $44.113 \pm 0.012$ | $72.567 \pm 0.040$ | $32.070 \pm 0.050$ |
| SimCLR | $77.300 \pm 0.014$ | $48.845 \pm 0.615$ | $75.533 \pm 0.026$ | $35.108 \pm 0.453$ |
|     + *margins* | $76.377 \pm 0.301$ | $47.023 \pm 0.219$ | $75.617 \pm 0.273$ | $34.747 \pm 0.430$ |
|     + *pos.* | $76.893 \pm 0.220$ | $49.090 \pm 0.262$ | $75.675 \pm 0.194$ | $35.067 \pm 0.380$ |
|     + *pos. & curv.* | $76.620 \pm 0.190$ | $48.683 \pm 0.107$ | $75.571 \pm 0.220$ | $34.280 \pm 0.215$ |
|     + *ratio.* | $\mathbf{77.440 \pm 0.346}$ | $\mathbf{49.903 \pm 0.025}$ | $\mathbf{75.983 \pm 0.029}$ | $\mathbf{35.237 \pm 0.032}$ |
| BYOL | $\mathbf{75.086 \pm 0.156}$ | $\mathbf{41.560 \pm 0.321}$ | $69.433 \pm 0.272$ | $\mathbf{30.306 \pm 0.309}$ |
|     + *curv.* | $74.820 \pm 0.552$ | $39.445 \pm 0.078$ | $\mathbf{72.442 \pm 0.366}$ | $29.260 \pm 0.028$ |

Tabs. 2 and 3 present the linear probing and transfer learning performance on four datasets. In addition, Tab. 4 shows the linear probing performance on ImageNet. As indicated in Tab. 2, the most significant improvement of MoCo occurs when positive samples are emphasized. The transfer learning performance (Tab. 3) demonstrates that this performance improvement is not due to overfitting but rather reflects enhanced representations capable of generalizing to unseen datasets. While $curv.$ coupled with $pos.$ can improve performance in seen datasets, it does not consistently enhance performance in unseen datasets. As mentioned in Sec. 5.2, emphasizing positive samples does not significantly improve SimCLR on four datasets. On ImageNet (Tab. 4), however, SimCLR shows improvement, and we believe this pertains to the increased batch size.

Scaling gradients by the ratio consistently improves contrastive learning across various datasets, not only in seen datasets but also in unseen datasets. The performance improvement is more apparent for SimCLR. BYOL is structurally less affected by margins or their associated effects. Consequently, performance improvements are limited. In conclusion, optimizing margins and related effects can contribute to performance enhancements on the target dataset, and emphasizing positive samples and scaling by the ratio appear to be important for achieving improved representations that generalize to unseen datasets.

## 6   LIMITATIONS AND DISCUSSION

Our work is based on several assumptions; cosine similarity and one-hot target probability $p_{ij}$. Since not all methods follow these assumptions, our work requires further validation to determine whether the observations made in this study can be transferred to other contrastive learning methods that violate these assumptions Chuang et al. (2020); Zheng et al. (2021). Moreover, we could not delve into the ratio $(p_{ij} - \beta q_{ij})/(p_{ij} - \beta \tilde{q}_{ij})$ due to the fact that it is not a separable function. Considering the improvement in SimCLR, delving into it can be an interesting direction.

## 7   CONCLUSION

We proposed a novel view on understanding the role of margins using gradient analysis and not relying on decision-boundary-based explanations. By analyzing gradients, we discovered that margins have a mixture of four different effects: emphasizing positive samples, weighting positive samples differently based on their angles, scaling gradients by the ratio of sums of exponentiated logits, and alleviating the diminishing gradient effect as the estimated probability approaches the target probability. We separated each effect and experimentally demonstrated the significance and limits of each. We hope our analysis of how margins affect the gradients of representation learning will help improve contrastive learning and possibly margins themselves.

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

# A PROOFS

## A.1 THEOREM 4.1

Recalling that $\tilde{\mathcal{L}}_i = -\sum_{j \in \mathcal{X}} p_{ij} \tilde{\delta}_{ij} + \beta \sum_{j \in \mathcal{X}} p_{ij} \log \sum_{k \in \mathcal{X}} \exp(\tilde{\delta}_{ik})$, the derivative with respect to $\tilde{\delta}_{ij}$ can be expressed as follows:

$$\frac{\partial \tilde{\mathcal{L}}_i}{\partial \tilde{\delta}_{ij}} = -p_{ij} + \beta \frac{\exp(\tilde{\delta}_{ij})}{\sum_{k \in \mathcal{X}} \exp(\tilde{\delta}_{ik})} = -p_{ij} + \beta \tilde{q}_{ij}. \tag{13}$$

Similarly, $\partial \tilde{\mathcal{L}}_i / \partial \delta_{ij}$ equals $-p_{ij} + \beta q_{ij}$. The derivative of $\tilde{\mathcal{L}}_i$ (Eq. 3) and $\mathcal{L}_i$ (Eq. 5) with respect to $\theta_{ij}$ can be represented as follows:

$$\frac{\partial \tilde{\mathcal{L}}_i}{\partial \theta_{ij}} = \frac{\partial \tilde{\mathcal{L}}_i}{\partial \tilde{\delta}_{ij}} \frac{\partial \tilde{\delta}_{ij}}{\partial \theta_{ij}} = (p_{ij} - \beta \tilde{q}_{ij}) \frac{\sin(\theta_{ij})}{\tau} \tag{14}$$

$$\frac{\partial \mathcal{L}_i}{\partial \theta_{ij}} = \frac{\partial \mathcal{L}_i}{\partial \delta_{ij}} \frac{\partial \delta_{ij}}{\partial \theta_{ij}} = (p_{ij} - \beta q_{ij}) \frac{\sin(\theta_{ij} + m_1 p_{ij})}{\tau} \tag{15}$$

Using Eqs. 14 and 15, we can draw the relationships between two derivatives (Theorem 4.1).

$$\frac{\partial \mathcal{L}_i}{\partial \theta_{ij}} = (p_{ij} - \beta q_{ij}) \frac{\sin(\theta_{ij} + m_1 p_{ij})}{\tau} \frac{p_{ij} - \beta \tilde{q}_{ij}}{p_{ij} - \beta \tilde{q}_{ij}} \frac{\sin(\theta_{ij})}{\sin(\theta_{ij})} \tag{16}$$

$$= (p_{ij} - \beta \tilde{q}_{ij}) \frac{\sin(\theta_{ij})}{\tau} \frac{p_{ij} - \beta q_{ij}}{p_{ij} - \beta \tilde{q}_{ij}} \frac{\sin(\theta_{ij} + m_1 p_{ij})}{\sin(\theta_{ij})} = \frac{\partial \tilde{\mathcal{L}}_i}{\partial \tilde{\delta}_{ij}} \frac{p_{ij} - \beta q_{ij}}{p_{ij} - \beta \tilde{q}_{ij}} \frac{\sin(\theta_{ij} + m_1 p_{ij})}{\sin(\theta_{ij})} \tag{17}$$

## A.2 LEMMA 4.2

We first assume that $\beta$ is one and $p_{ij}$ is either zero or one. Before proving the lemman, we first reformulate $\tilde{q}_{ij}$ and $q_{ij}$ as follows:

$$\tilde{q}_{ij} = \frac{\exp(\tilde{\delta}_{ij})}{\sum_{k \in \mathcal{X}} \exp(\tilde{\delta}_{ik})} = \frac{\exp(\tilde{\delta}_{ij})}{\exp(\tilde{\delta}_{il})/\tilde{q}_{il}} \tag{18}$$

$$q_{ij} = \frac{\exp(\delta_{ij})}{\sum_{k \in \mathcal{X}} \exp(\delta_{ik})} = \frac{\exp(\delta_{ij})}{\exp(\delta_{il}) - \exp(\tilde{\delta}_{il}) + \sum_{k \in \mathcal{X}} \exp(\tilde{\delta}_{ik})} \tag{19}$$

$$= \frac{\exp(\delta_{ij})}{\exp(\delta_{il}) - \exp(\tilde{\delta}_{il}) + \exp(\tilde{\delta}_{il})/\tilde{q}_{il}} \tag{20}$$

$$= \frac{\exp(\delta_{ij})}{\exp(\delta_{il}) + \exp(\tilde{\delta}_{il})(1/\tilde{q}_{il} - 1)}. \tag{21}$$

Using these reformulations, we can express $(p_{ij} - \beta q_{ij})/(p_{ij} - \beta \tilde{q}_{ij})$ as follows:

$$\frac{p_{ij} - \beta q_{ij}}{p_{ij} - \beta \tilde{q}_{ij}} = \frac{(p_{ij}(\exp(\delta_{il}) + \exp(\tilde{\delta}_{il})(1/\tilde{q}_{il} - 1)) - \beta \exp(\delta_{ij}))/(\exp(\delta_{il}) + \exp(\tilde{\delta}_{il})(1/\tilde{q}_{il} - 1))}{(p_{ij} \exp(\tilde{\delta}_{il})/\tilde{q}_{il} - \beta \exp(\tilde{\delta}_{ij}))/(\exp(\tilde{\delta}_{il})/\tilde{q}_{il})} \tag{22}$$

$$= \frac{p_{ij}(\exp(\delta_{il}) + \exp(\tilde{\delta}_{il})(1/\tilde{q}_{il} - 1)) - \beta \exp(\delta_{ij})}{p_{ij} \exp(\tilde{\delta}_{il})/\tilde{q}_{il} - \beta \exp(\tilde{\delta}_{ij})} \frac{\exp(\tilde{\delta}_{il})/\tilde{q}_{il}}{\exp(\delta_{il}) + \exp(\tilde{\delta}_{il})(1/\tilde{q}_{il} - 1)} \tag{23}$$

For a positive sample $l$,

$$\frac{p_{il} - \beta q_{il}}{p_{il} - \beta \tilde{q}_{il}} = \frac{\exp(\delta_{il}) + \exp(\tilde{\delta}_{il})(1/\tilde{q}_{il} - 1) - \exp(\delta_{ij})}{\exp(\tilde{\delta}_{il})/\tilde{q}_{il} - \exp(\tilde{\delta}_{ij})} \frac{\exp(\tilde{\delta}_{il})/\tilde{q}_{il}}{\exp(\delta_{il}) + \exp(\tilde{\delta}_{il})(1/\tilde{q}_{il} - 1)} \tag{24}$$

$$= \frac{\exp(\tilde{\delta}_{il})/\tilde{q}_{il} - \exp(\tilde{\delta}_{ij})}{\exp(\tilde{\delta}_{il})/\tilde{q}_{il} - \exp(\tilde{\delta}_{ij})} \frac{\exp(\tilde{\delta}_{il})/\tilde{q}_{il}}{\exp(\delta_{il}) + \exp(\tilde{\delta}_{il})(1/\tilde{q}_{il} - 1)} \tag{25}$$

$$= \frac{\exp(\tilde{\delta}_{il})/\tilde{q}_{il}}{\exp(\delta_{il}) + \exp(\tilde{\delta}_{il})(1/\tilde{q}_{il} - 1)}. \tag{26}$$

Using the fact that $\delta_{ih}$ equals $\tilde{\delta}_{ih}$ for a negative sample $h$, we can get the following equation:

$$\frac{p_{ih} - \beta q_{ih}}{p_{ih} - \beta\tilde{q}_{ih}} = \frac{-\exp(\delta_{ih})}{-\exp(\tilde{\delta}_{ih})} \frac{\exp(\tilde{\delta}_{il})/\tilde{q}_{il}}{\exp(\delta_{il}) + \exp(\tilde{\delta}_{il})(1/\tilde{q}_{il} - 1)} = \frac{\exp(\tilde{\delta}_{il})/\tilde{q}_{il}}{\exp(\delta_{il}) + \exp(\tilde{\delta}_{il})(1/\tilde{q}_{il} - 1)}$$

$$(27)$$

That is, $(p_{ij} - \beta q_{ij})/(p_{ij} - \beta\tilde{q}_{ij})$ equals $(\exp(\tilde{\delta}_{il})/q_{il}) / (\exp(\delta_{il}) + \exp(\tilde{\delta}_{il})(1/q_{il} - 1))$, and it can also be expressed as $\sum \exp(\tilde{\delta}_{ij})/\sum \exp(\delta_{ij})$.

### A.3 LEMMA 4.3

We first assume that there are only one positive sample and other samples are negative. This assumption holds true for SSL methods we are analyzing. In addition, we also assume that $\beta$ to one (which holds true for MoCo and SimCLR). If $\beta$ is zero, then subtractive margin $m_2$ cannot work. Eq. 15 can be rewritten as follows:

$$\frac{\partial \mathcal{L}_i}{\partial \theta_{ij}} = (p_{ij} - \beta q_{ij})\frac{\sin(\theta_{ij} + m_1 p_{ij})}{\tau} = \frac{-\beta \exp(\delta_{ij}) + p_{ij}\sum_{k\in\mathcal{X}}\exp(\delta_{ik})}{\sum_{k\in\mathcal{X}}\exp(\delta_{ik})}\frac{\sin(\theta_{ij} + m_1 p_{ij})}{\tau}$$

$$(28)$$

$$= \frac{-\beta\exp(\delta_{ij}) + p_{ij}\sum_{k\in\mathcal{X}}\exp(\delta_{ik})}{\sum_{k\in\mathcal{X}}\exp(\tilde{\delta}_{ik})}\frac{\sum_{k\in\mathcal{X}}\exp(\tilde{\delta}_{ik})}{\sum_{k\in\mathcal{X}}\exp(\delta_{ik})}\frac{\sin(\theta_{ij} + m_1 p_{ij})}{\tau}.$$

$$(29)$$

In addition, we reformualte $\exp \delta_{ij}$ as follows,

$$exp(\delta_{ij}) = \exp(\frac{\cos(\theta_{ij})}{\tau})\exp(\frac{\cos(\theta_{ij} + m_1 p_{ij}) - \cos(\theta_{ij}) - m_2 p_{ij}}{\tau}).$$

$$(30)$$

$$= exp(\tilde{\delta}_{ij})\exp(\frac{\cos(\theta_{ij} + m_1 p_{ij}) - \cos(\theta_{ij}) - m_2 p_{ij}}{\tau})$$

$$(31)$$

For brevity, we will use the notation $\eta_{ij}$ for $\exp((\cos(\theta_{ij} + m_1 p_{ij}) - \cos(\theta_{ij}) - m_2 p_{ij})/\tau)$. We can rewrite the derivative as follows;

$$\frac{\partial \mathcal{L}_i}{\partial \theta_{ij}} = \frac{-\beta\exp(\tilde{\delta}_{ij})\eta_{ij} + p_{ij}\sum_{k\in\mathcal{X}}\exp(\tilde{\delta}_{ik})\eta_{ik}}{\sum_{k\in\mathcal{X}}\exp(\tilde{\delta}_{ik})}\frac{\sum_{k\in\mathcal{X}}\exp(\tilde{\delta}_{ik})}{\sum_{k\in\mathcal{X}}\exp(\tilde{\delta}_{ik})\eta_{ik}}\frac{\sin(\theta_{ij} + m_1 p_{ij})}{\tau}.$$

$$(32)$$

We will use $l$ to denote the index of a positive sample. Furthermore, we use the fact that $\eta_{ih}$ of negative sample $h$ equals one. We can further simplify the equation using this assumption as follows;

$$\frac{\partial \mathcal{L}_i}{\partial \theta_{ij}} = \frac{-\beta\exp(\tilde{\delta}_{ij})\eta_{ij} + p_{ij}(\exp(\tilde{\delta}_{il})(\eta_{il} - 1) + \sum_{k\in\mathcal{X}}\exp(\tilde{\delta}_{ik}))}{\sum_{k\in\mathcal{X}}\exp(\tilde{\delta}_{ik})}\frac{\sum_{k\in\mathcal{X}}\exp(\tilde{\delta}_{ik})}{\sum_{k\in\mathcal{X}}\exp(\tilde{\delta}_{ik})\eta_{ij}}\frac{\sin(\theta_{ij} + m_1 p_{ij})}{\tau}$$

$$(33)$$

$$= (p_{ij} + \frac{-\beta\exp(\tilde{\delta}_{ij})\eta_{ij} + p_{ij}\exp(\tilde{\delta}_{il})(\eta_{il} - 1)}{\sum_{k\in\mathcal{X}}\exp(\tilde{\delta}_{ik})})\frac{\sum_{k\in\mathcal{X}}\exp(\tilde{\delta}_{ik})}{\sum_{k\in\mathcal{X}}\exp(\tilde{\delta}_{ik})\eta_{ij}}\frac{\sin(\theta_{ij} + m_1 p_{ij})}{\tau}$$

$$(34)$$

$$= (p_{ij} - \beta\tilde{q}_{ij}\eta_{ij} + p_{ij}\tilde{q}_{il}(\eta_{il} - 1))\frac{\sum_{k\in\mathcal{X}}\exp(\tilde{\delta}_{ik})}{\sum_{k\in\mathcal{X}}\exp(\tilde{\delta}_{ik})\eta_{ij}}\frac{\sin(\theta_{ij} + m_1 p_{ij})}{\tau}$$

$$(35)$$

$$= (p_{ij} - \beta\tilde{q}_{ij}\eta_{ij} + p_{ij}\tilde{q}_{il}(\eta_{il} - 1))\frac{\sum_{k\in\mathcal{X}}\exp(\tilde{\delta}_{ik})}{\exp(\tilde{\delta}_{il})(\eta_{il} - 1) + \sum_{k\in\mathcal{X}}\exp(\tilde{\delta}_{ik})}\frac{\sin(\theta_{ij} + m_1 p_{ij})}{\tau}$$

$$(36)$$

$$= (p_{ij} - \beta\tilde{q}_{ij}\eta_{ij} + p_{ij}\tilde{q}_{il}(\eta_{il} - 1))\frac{1}{1 - (1 - \eta_{il})\tilde{q}_{il}}\frac{\sin(\theta_{ij} + m_1 p_{ij})}{\tau}.$$

$$(37)$$

Since $\beta$ to one by the assumption, gradients for a positive sample $l$ can rewritten as follows;

$$\frac{\partial \mathcal{L}_i}{\partial \theta_{il}} = (p_{il} + \tilde{q}_{il}(p_{il}(\eta_{il} - 1) - \beta\eta_{il}))\frac{1}{1 - (1 - \eta_{il})\tilde{q}_{il}}\frac{\sin(\theta_{il} + m_1 p_{il})}{\tau} \tag{38}$$

$$= (p_{il} - \tilde{q}_{il})\frac{\sin(\theta_{il} + m_1 p_{il})}{\tau}\frac{1}{1 - (1 - \eta_{il})\tilde{q}_{il}}. \tag{39}$$

Similarly, by using $\eta_{ih}$ equals one, we can get the gradients for a negative sample $h$ as follows;

$$\frac{\partial \mathcal{L}_i}{\partial \theta_{ih}} = (p_{ih} - \beta\tilde{q}_{ih}\eta_{ih})\frac{1}{1 - (1 - \eta_{il})\tilde{q}_{il}}\frac{\sin(\theta_{ih} + m_1 p_{ih})}{\tau} \tag{40}$$

$$= (p_{ih} - \tilde{q}_{ih})\frac{\sin(\theta_{ih} + m_1 p_{ih})}{\tau}\frac{1}{1 - (1 - \eta_{il})\tilde{q}_{il}}. \tag{41}$$

That is, under the above assumptions, subtractive margin multiplies both positive and negative sample gradients by $1/(1 - (1 - \eta_{il})\tilde{q}_{il})$. If we reformulate the equation, we get the role of the subtractive margins:

$$\frac{\partial \mathcal{L}_i}{\partial \theta_{ij}} = (p_{ij} - \tilde{q}_{ij})\frac{\sin(\theta_{ij} + m_1 p_{ij})}{\tau}\frac{1}{1 - (1 - \eta_{il})\tilde{q}_{il}} \tag{42}$$

$$= (p_{ij} - \tilde{q}_{ij})\frac{\sin(\theta_{ij})}{\sin(\theta_{ij})}\frac{\sin(\theta_{ij} + m_1 p_{ij})}{\tau}\frac{1}{1 - (1 - \eta_{il})\tilde{q}_{il}} \tag{43}$$

$$= \frac{\partial \tilde{\mathcal{L}}_i}{\partial \theta_{ij}}\frac{\sin(\theta_{ij} + m_1 p_{ij})}{\sin(\theta_{ij})}\frac{1}{1 - (1 - \exp((\cos(\theta_{il} + m_1) - \cos(\theta_{il}) - m_2)/\tau))\tilde{q}_{il}}. \tag{44}$$

### A.4  LEMMA 4.4

As shown in Eq. 41, $\partial \mathcal{L}_i / \partial \theta_{ij}$ can be expressed as follows:

$$\frac{\partial \mathcal{L}_i}{\partial \theta_{ij}} = (p_{ij} - \tilde{q}_{ij})\frac{\sin(\theta_{ij} + m_1 p_{ij})}{\tau}\frac{1}{1 - (1 - \exp((\cos(\theta_{il} + m_1) - \cos(\theta_{il}) - m_2)/\tau))\tilde{q}_{il}}. \tag{45}$$

Therefore, we can get the following equation for a positive sample $l$:

$$\lim_{m_2 \to \infty}\frac{\partial \mathcal{L}_i}{\partial \theta_{il}} = \lim_{m_2 \to \infty}(p_{il} - \tilde{q}_{il})\frac{\sin(\theta_{il} + m_1 p_{il})}{\tau(1 - (1 - \exp((\cos(\theta_{il} + m_1) - \cos(\theta_{il}) - m_2)/\tau))\tilde{q}_{il})} \tag{46}$$

$$= (p_{il} - \tilde{q}_{il})\frac{\sin(\theta_{il} + m_1 p_{il})}{(1 - \tilde{q}_{il})\tau} = \frac{\sin(\theta_{il} + m_1)}{\tau}. \tag{47}$$

Table 5: Margin values for four datasets (Tab. 2).

| Method | Dataset | $m1$ | $m2$ |
|--------|---------|------|------|
| *MoCov3* | CIFAR10 | 0.1 | 0.4 |
| | CIFAR100 | 0.5 | 0.7 |
| | STL10 | 0.4 | 0.6 |
| | TinyImageNet | 0.1 | 0.4 |
| *SimCLR* | CIFAR10 | 0.5 | 0.4 |
| | CIFAR100 | 0.6 | 0.6 |
| | STL10 | 0.0 | 0.2 |
| | TinyImageNet | 0.0 | 0.8 |

Table 6: Scaling factors of *pos.* only models.

| Method | Dataset | *pos.* ($s$) |
|--------|---------|------|
| *MoCov3* | CIFAR10 | 20 |
| | CIFAR100 | 20 |
| | STL10 | 40 |
| | TinyImageNet | 20 |
| *SimCLR* | CIFAR10 | 1.125 |
| | CIFAR100 | 1.0625 |
| | STL10 | 1.0625 |
| | TinyImageNet | 1.0625 |

Table 7: Scaling factors and curvature factors.

| Method | Dataset | *pos.* ($s$) | *curv.* ($c$) |
|--------|---------|------|------|
| *MoCov3* | CIFAR10 | 20 | 0.7 |
| | CIFAR100 | 40 | 1 |
| | STL10 | 40 | 2.5 |
| | TinyImageNet | 40 | 2.5 |
| *SimCLR* | CIFAR10 | 2.5 | 0.7 |
| | CIFAR100 | 2 | 0.7 |
| | STL10 | 1.25 | 0.7 |
| | TinyImageNet | 1.125 | 1.0 |
| *BYOL* | CIFAR10 | - | 1.0 |
| | CIFAR100 | - | 2.5 |
| | STL10 | - | 5.0 |
| | TinyImageNet | - | 2.5 |

Table 8: Angular margin $m_1$ to calculate the ratio (Tabs. 2 and 3)

| Method | Dataset | $m_1$ |
|--------|---------|------|
| *MoCov3* | CIFAR10 | 1.6 |
| | CIFAR100 | 1.6 |
| | STL10 | 0.4 |
| | TinyImageNet | 1.6 |
| *SimCLR* | CIFAR10 | 0.2 |
| | CIFAR100 | 0.2 |
| | STL10 | 0.4 |
| | TinyImageNet | 0.2 |

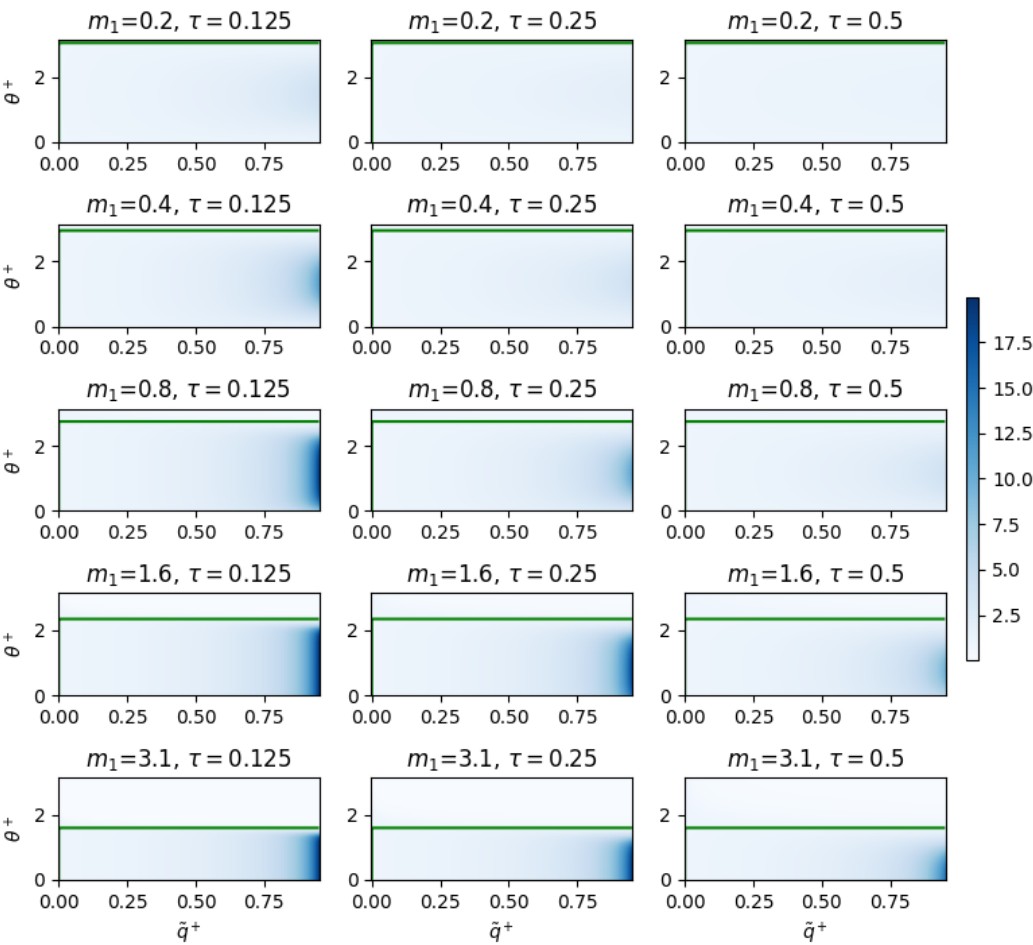

Figure 8: Visualization of $(p_{ij} - \beta q_{ij})/(p_{ij} - \beta \tilde{q}_{ij})$. The green lines indicate where the ratio is one. $m_1$ and $\tau$ are the angular margin and temperature, respectively (Eq. 4).

## B  VISUALIZATION

Fig. 8 visualizes the weight distributions of the ratios $(p_{ij} - \beta q_{ij})/(p_{ij} - \beta \tilde{q}_{ij})$. Angular margin $m_1$ and temperature $\tau$ are from Eq. 4. The figure shows that as $m_1$ increases, the center of emphasis in $\theta^+$ lowers and the emphasizing area broadens.

## C  IMPLEMEMTATION DETAILS

### C.1  EXPERIMENTAL SETTINGS FOR CIFAR-10, CIFAR-100, STL-10, AND TINYIMAGENET

Our experimental settings for these dataset basically follows that of ReSSL. We pretrained each model on a source dataset for 200 epochs and then fine-tuned it on a target dataset for 100 epochs. We set the batch size to 256 and $\tau$ to 0.25 (for MoCov3 and SimCLR). The latent feature dimension was set to 128, while the hidden dimension of linear layers (including projection heads and predictors) was set to 2048. We used SGD with a momentum of 0.9 for both pretraining and evaluation. The learning rate was set to 0.06 for pretraining the backbone model and 1 for fine-tuning the linear layer. During evaluation, we employed the cosine annealing learning rate scheduler.

We used two augmentation policies: strong and weak augmentation. The strong augmentation consists of random resize cropping, horizontal flipping, color jittering, random gray scaling, and Gaussian blurring. In contrast, the weak augmentation only included random horizontal and random cropping. For contrastive learning-based SSL methods with a teacher model (MoCov3, BYOL), we used weak

data augmentation to the teacher model and strong data augmentation to the student model. For a method without a teacher model (SimCLR), however, we use these two different augmentation policies to generate two different representations of the same identity.

For projection heads and predictors of MoCov3, SimCLR, and BYOL, we used batch normalization before the ReLU activation.

During hyperparameter tuning for comparison with baselines, we only adjusted margins ($m_1$ and $m_2$), $s$ (Eq.9), and $c$ (Eq.11), while keeping other hyperparameters fixed, such as the learning rate. The hyperparameters used in Sec. 5.6 are specified in Tabs. 5, 8 and 6.

### C.2 Experimental settings for ImageNet

We used the official implementations of both MoCov3 and SimCLR. Due to our computational budget, we pretrained the networks for 100 epochs. Following the experimental settings of MoCov3 and SimCLR, we fine-tuned the last linear layer for 90 epochs for evaluation. We only modified the gradient scales using Alg. 3, without making any other changes. When training SimCLR, we used a two-linear layered projection module. This choice was made because only the official performance of the two-linear-layered version, pretrained for 100 epochs, was available. For both MoCo and SimCLR, we use a batch size of 4,096.

The hyperparameters ($s$ and $c$), used in Tab. 4 are as follows: for MoCov3, we set $s$ to 10 and $c$ to 1.5, while for SimCLR, $s$ was set to 2 and $c$ to 1.

## D Additional Experiments

We ran additional experiments on MoCov3 for much longer epochs (1,000). Tabs. 9 and 10 show that emphasizing positive samples improves performance not only in the short run but even in the long run.

Table 9: Linear Probing of MoCov3.

| $s$ | epochs | CIFAR10 | CIFAR100 | STL10 | TinyImageNet |
|---|---|---|---|---|---|
| 1 | 1000 | $91.142 \pm 0.218$ | $65.010 \pm 0.215$ | $90.280 \pm 0.113$ | $45.682 \pm 0.151$ |
| 10 | 1000 | $92.386 \pm 0.100$ | $69.136 \pm 0.421$ | $90.633 \pm 0.244$ | $50.478 \pm 0.237$ |
| 20 | 1000 | $92.624 \pm 0.161$ | $\mathbf{69.466 \pm 0.183}$ | $90.860 \pm 0.122$ | $\mathbf{51.154 \pm 0.184}$ |
| 40 | 1000 | $\mathbf{92.648 \pm 0.171}$ | $69.408 \pm 0.342$ | $\mathbf{91.098 \pm 0.209}$ | $51.138 \pm 0.362$ |

Table 10: Transfer Learning of MoCov3.

| $s$ | epochs | CIFAR100 $\rightarrow$ CIFAR10 | CIFAR10 $\rightarrow$ CIFAR100 | TinyImageNet $\rightarrow$ STL10 | STL10 $\rightarrow$ TinyImageNet |
|---|---|---|---|---|---|
| 1 | 1000 | $75.298 \pm 0.310$ | $40.696 \pm 0.327$ | $71.088 \pm 0.361$ | $30.358 \pm 0.304$ |
| 10 | 1000 | $79.858 \pm 0.255$ | $53.028 \pm 0.569$ | $77.533 \pm 0.101$ | $38.702 \pm 0.420$ |
| 20 | 1000 | $\mathbf{80.144 \pm 0.105}$ | $54.515 \pm 0.235$ | $78.623 \pm 0.193$ | $40.464 \pm 0.341$ |
| 40 | 1000 | $79.800 \pm 0.162$ | $\mathbf{54.612 \pm 0.317}$ | $\mathbf{78.810 \pm 0.198}$ | $\mathbf{40.600 \pm 0.360}$ |

## E Pseudo code

In this section, we provide pseudo codes for reproduction. Although contrastive learning-based baselines do not explicitly utilize angles, as mentioned in the main paper, computing logits using cosine similarity can be interpreted as obtaining logits using the angles that result from taking the arccos of the cosine similarity values (Alg. 1). We used Alg. 2 to use margins in contrastive learning methods. To scale the gradients, it can be accomplished by replacing only the process of converting

angles to logits from Alg. 1 to Algs. 3, 4, 5, and 6. As the algorithms demonstrate, our modification does not affect the logit values but solely modifies the gradients. To adjust the curvature of positive gradient scales in BYOL, it suffices to set $p$ as an all-ones vector since only positive samples are used.

---

**Algorithm 1** Logits as a function of angles: pseudo code of general contrastive learning (Eq. 2)

---

**Inputs:**
    $\theta$      Angles
    $\tau$      Temperature
$logits \leftarrow \cos(\theta)/\tau$
**return** $logits$

---

---

**Algorithm 2** Logits with margins ($m_1$ and $m_2$) (Eq. 4)

---

**Inputs:**
    $\theta$      Angles
    $\tau$      Temperature
    $p$      Target probabilities
    $m_1$    Angular margin
    $m_2$    Subtractive margin
$logits \leftarrow (\cos(\theta + p \times m_1) - p \times m_2)/\tau$
**return** $logits$

---

---

**Algorithm 3** Emphasizing positive samples and controlling the curvature of positive gradient scales (Fig. 6, Tabs. 4 and 2).

---

**Inputs:**
    $\theta$      Angles
    $\tau$      Temperature
    $p$      Target probabilities
    $s$      Scaling factor
    $c$      Curvature factor
$logits \leftarrow \cos(\theta)/\tau$
$scales \leftarrow stop\_gradient(s \times (1 - \frac{\theta}{\pi}^c)^{1/c})$
$logits \leftarrow (1 - p) \times logits + p \times ((1 - scales) \times stop\_gradient(logits) + scales \times logits)$
**return** $logits$

---

---

**Algorithm 4** Scaling gradients by the ratio (Fig. 7).

---

**Inputs:**

| | |
|---|---|
| $\theta$ | Angles |
| $\tau$ | Temperature |
| $p$ | Target probabilities |
| $m_1$ | Angular margin |

$logits \leftarrow \cos(\theta)/\tau$
$new\_logits \leftarrow \cos(\theta + p \times m_1)/\tau$
$ratio \leftarrow sum(\exp(logits), dim = -1)/sum(\exp(new\_logits), dim = -1)$
$scales \leftarrow stop\_gradient(ratio)$
$logits \leftarrow (1 - p) \times logits + p \times ((1 - scales) \times stop\_gradient(logits) + scales \times logits)$
**return** $logits$

---

**Algorithm 5** Attenuating the diminishing gradients (Type I) (Eq. 12)

---

**Inputs:**

| | |
|---|---|
| $\theta$ | Angles |
| $\tau$ | Temperature |
| $p$ | Target probabilities |
| $\alpha$ | Attenuation factor |

$logits \leftarrow \cos(\theta)/\tau$
$q \leftarrow softmax(logits)$
$scales \leftarrow sum(p/stop\_gradient(p - \alpha \times q), \ dim = -1)$
$logits \leftarrow (1 - scales) \times stop\_gradient(logits) + scales \times logits$
**return** $logits$

---

**Algorithm 6** Attenuating the diminishing gradients of positive samples (Type II) (Eq. 12)

---

**Inputs:**

| | |
|---|---|
| $\theta$ | Angles |
| $\tau$ | Temperature |
| $p$ | Target probabilities |
| $\alpha$ | Attenuation factor |

$logits \leftarrow \cos(\theta)/\tau$
$q \leftarrow softmax(logits)$
$scales \leftarrow 1/stop\_gradient(p - \alpha \times q)$
$logits \leftarrow (1 - p) \times logits + p \times ((1 - scales) \times stop\_gradient(logits) + scales \times logits)$
**return** $logits$

---

