# OpenReview forum: "Understanding Contrastive Learning Through the Lens of Margins"
_ICLR.cc/2024/Conference — ICLR 2024 Conference Withdrawn Submission_

### Official Review · Reviewer_Bk8n · 2023-10-29

**Soundness:** 3 good
**Presentation:** 3 good
**Contribution:** 3 good
**Rating:** 6
**Confidence:** 5

**Summary:**

This work focuses on the effect of margin in contrastive learning, which is similar to face recognition tasks. The authors find a mixture of effects of margins in contrastive learning through gradient analysis. Then they separate each effect to validate its effect and limitation. Finally, the authors propose to emphasize positive samples and scale gradients based on positive sample angles and logits to improve contrastive learning. They evaluate the effectiveness on multiple datasets and conduct extensive ablation studies.

**Strengths:**

1.The paper is organized and written well, and the motivation is clear to me.
2.The theoritical analysis is organized and comprehensive, and the analysis on margin is also useful for contrastive learning.
3.The authors conduct extensive experments and ablation studies to support their analysis.

**Weaknesses:**

1.There is a key issue not considered in the paper. In fact, I think an important difference between contrastive learning and face representation learning is that the negative pairs in contrastive learning may belong to the same real class due to lack of labels, i.e., fake negative problem. So using margin to overly push them apart will harm the representation learning. So how the authors think about that problem?
2.The proposed method is too simple and not systematic. Although the authors provide detail analysis on angular margin and subtractive margin, they did not provide any improved operations.
3.How hyper-parameter s scales positive sample gradients in Equation 8 and 9? These variables seem to have nothing to do with the gradient.

**Questions:**

Please respond to the weakness above.

---

### Official Review · Reviewer_Q3Bo · 2023-10-31

**Soundness:** 3 good
**Presentation:** 2 fair
**Contribution:** 2 fair
**Rating:** 5
**Confidence:** 3

**Summary:**

This paper proposes to understand the role of margins based on gradient analysis.

They analyze how the margins affect the gradients of contrastive learning and they separate the effect into more elemental levels.

The also analyze each and provide some directions to improve contrastive learning.

Their experimental results show that emphasizing positive samples and scaling the gradients depending on the positive sample angles and logits are the key to boosting the generalization of contrastive learning in both seen and unseen datasets.

**Strengths:**

+ Overall the paper is technical sound and well-written.

+ There are some nice visualizations and comparisons presented in the experimental section.

+ The paper deals with a challenging and interesting topic in contrastive learning, and gives insights to existing contrastive learning methods.

**Weaknesses:**

Major:

- Does the proposed method generalized to the Canonical Correlation Analysis family, e.g., Barlow Twins?

- It is not very clear to reviewer how the angular margin and subtractive margin are derived? The intuition is not very clearer to reviewer, could you please explain? Also why 'these margins are added only to positive samples' (as in Sec 3.2)?

- The in-text references are quite strange / messy.

- Some sentences, due to its longer nature, are hard to understand. For example, In the last paragraph of the Introduction section, 'We believe that understanding how margins affect contrastive learning can provide direction for ... Not only that, we believe our new perspective on margins could ...'

- The contributions listed in the Introduction section are too vague, e.g., the second contribution is not very clear to review, could you please explain it and make it clear?

- Related work section, the writing quality is very poor, e.g., MoCo He et al. SimCLR Chen et al, BYOL Grill et al, etc. It might because of the in-text reference issues again.

- In the related work section, the authors only presented some literature review, and the differences w.r.t. the existing works versus the proposed solution are not highlighted. This is a noteworthy shortcoming. Moreover, there are not any in-depth discussions w.r.t. the proposed solution.

- In the approach section, as a theoretical paper, it is suggested to have a notation section detailing the maths symbols used in the paper, e.g., for vectors, matrices, etc.

- The reviewer is confusing w.r.t. Sec 5.2 'The larger the batch size, the more the performance curve becomes similar to that of MoCo'.

- What is the baseline performance in Fig. 5 and 6, is it possible to show it in the figures as well? Also, the figures are a kind of unclearer enough due to it is a bit too tiny and cluttered.

- Are there any insights and discussions w.r.t. why the experimental results are not very consistent across different datasets as in Table 2 & 3. What are the factors that affect this behavior?

- Sec. 6, the limitations and discussion is a bit too limited. It does not convey useful insights to readers, would it possible to make it nicer and deliver some fruitful insights and discussions?


Minor:

- Figure 3, first row 4 figures, the x axis, numbers are nearly disappeared.

**Questions:**

Please refer to the weakness section for the concerns and questions to be addressed.

---

### Official Review · Reviewer_Ykxx · 2023-10-31

**Soundness:** 3 good
**Presentation:** 3 good
**Contribution:** 2 fair
**Rating:** 5
**Confidence:** 5

**Summary:**

In this paper, the authors try to replace infoNCE with large-margin softmax, which has been widely used in metric learning. Based on this, the authors give gradient analysis and comparison with other baselines.

**Strengths:**

1. It is easy to follow.

**Weaknesses:**

1. The idea is not new for contrastive learning. The NLP community has investigated this before, and the CV (Max-Margin Contrastive Learning,AAAI2022) also has a similar work.

2. And it cannot make any insightful points like 'large-margin contrastive learning with distance polarization regularizer, ICML 2021'. Replacing infoNCE with large-margin softmax directly cannot help us understand contrastive learning. Because in large-margin softmax, we do not have false negative pair but in contrastive learning we have. So the only reason large-margin softmax will help us in CL should only be semi-hard mining mechanisim.

3. The gradient analysis is trivial. Because it cannot provide any insightful point and prove it (please check "On The Inadequacy of Optimizing Alignment and Uniformity in Contrastive Learning of Sentence Representations, ICLR2023"). At least this paper provides a point based on the gradient analysis.

4. This paper lost so many important citations and has no comparison with the related methods of contrastive learning with margin. For example, the large-margin softmax should be from "large-margin softmax loss for convolutional neural networks, ICML2016".

**Questions:**

Why the margin can help contrastive learning? Is there any insightful point?

---

### Official Review · Reviewer_Rz4Z · 2023-11-03

**Soundness:** 2 fair
**Presentation:** 1 poor
**Contribution:** 1 poor
**Rating:** 3
**Confidence:** 3

**Summary:**

This paper considers adding margin term in the conventional contrastive loss and check the effect of margin. The authors analyze  the behavior of gradient when margin term is introduced. The authors claim that the theoretical results on gradient are supported by empirical results.

**Strengths:**

* The approach is novel
* The authors ran several experiments with good ablation.

**Weaknesses:**

* Shallow theoretical depth
* Hard to get some insight on what Fig.1,2 and the theoretical results in Sec.4 imply
* The authors claim that 4 effects of introducing the margin term are empirically are covered in Sec.5. First, it was hard for me to get those 4 messages from both theory (in Sec.4) and in experiments (in Sec.5). Let us say 4 effects are observed. Does it explain why introducing the margin is beneficial for contrastive learning?
* Probably many of these issues are from the quality of writing.

**Questions:**

* In equation (3), it is unclear how the \beta term in the denominator (inside the log) is coming out to the linear form in the righthand side. Probably have typo somewhere? When \beta=0 (i.e., when only the positive samples are considered), the denominator inside the log becomes 0, which is also weird.